# NatureLM-audio: an Audio-Language Foundation Model for Bioacoustics

**David Robinson,**[*] **Marius Miron, Masato Hagiwara, Olivier Pietquin**
Earth Species Project

## Abstract

Large language models (LLMs) prompted with text and audio have achieved state-of-the-art performance across various auditory tasks, including speech, music, and general audio, showing emergent abilities on unseen tasks. However, their potential has yet to be fully demonstrated in bioacoustics tasks, such as detecting animal vocalizations in large recordings, classifying rare and endangered species, and labeling context and behavior—tasks that are crucial for conservation, biodiversity monitoring, and animal behavior studies. In this work, we present NatureLM-audio, the first audio-language foundation model specifically designed for bioacoustics. Our training dataset consists of carefully curated text-audio pairs spanning bioacoustics, speech, and music, designed to address the field's limited availability of annotated data. We demonstrate successful transfer of learned representations from music and speech to bioacoustics, and our model shows promising generalization to unseen taxa and tasks. We evaluate NatureLM-audio on a novel benchmark (BEANS-Zero) and it sets a new state of the art on several bioacoustics tasks, including zero-shot classification of unseen species. To advance bioacoustics research, we release our model weights, benchmark data, and open-source the code for training and benchmark data generation and model training. [1]

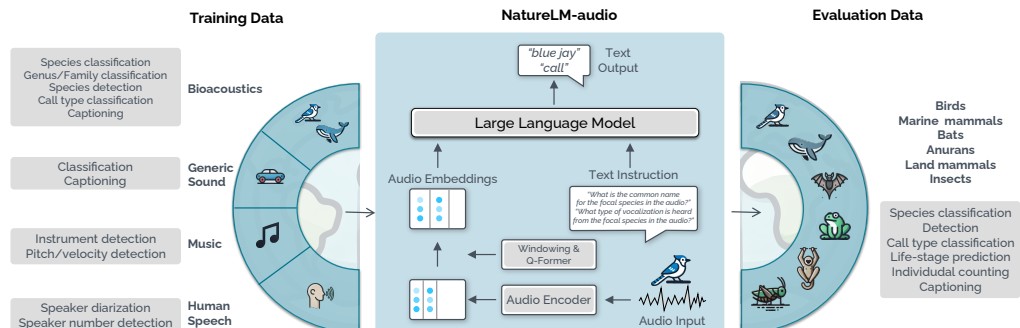

Figure 1: Overview of NatureLM-audio.

# 1 Introduction

Bioacoustics, the study of sound production and reception in animals, aims to understand animal behavior (Fischer et al., 2013), monitor biodiversity (Stowell, 2022), and model the mechanisms underlying animal communication (Bradbury & Vehrencamp, 1998). It plays a vital role in conservation and ecological research, as animal vocalizations provide key insights into ecosystem health, species interactions, and population dynamics. By enabling the detection of endangered species and the tracking of migration patterns, bioacoustic research directly contributes to biodiversity monitoring and conservation efforts (Rutz et al., 2023; Stevens et al., 2024).

---

[*]Corresponding author. `david@earthspecies.org`

[1]Project page: `https://earthspecies.github.io/naturelm-audio-demo/`

In recent years, machine learning has taken on an increasingly pivotal role in bioacoustic research. Beyond its role in large-scale ecological monitoring, it has opened up new frontiers in the study of animal communication, enabling discoveries such as the use of specialized vocalizations to label conspecifics in marmosets (Oren et al., 2024), dolphins (King & Janik, 2013), and elephants (Pardo et al., 2024). However, due to inherent challenges in data collection and annotation, many of these studies rely on strongly labeled small datasets (Stowell, 2022) and require careful statistical analysis to ensure significance and mitigate over-fitting. Meanwhile, vast amounts of unannotated bioacoustics data are recorded daily, particularly through passive acoustic monitoring (PAM, Dufourq et al. (2021)) and citizen science platforms such as Xeno-canto (Vellinga & Planqué, 2015). This growing data availability underscores the need for machine learning models capable of large-scale detection, classification, and annotation. The recent successes of large scale AI models in various domains—including natural language processing, computer vision, and game-playing—suggests the possibility of leveraging these large, raw datasets to learn robust and generalizable representations for bioacoustics (Ghani et al., 2023; Boudiaf et al., 2023).

Existing bioacoustics machine learning models are typically designed for specific species or tasks (Dufourq et al., 2021; Kahl et al., 2021; Cauzinille et al., 2024), limiting their generalizability. Many traditional approaches rely on small datasets focusing on a few species and individuals, validating results through statistical measures despite the risks of over-fitting. More recent models, such as BirdNET (Kahl et al., 2021) and Perch (Ghani et al., 2023), achieve strong performance in bird classification but require training dedicated classifiers for each target taxon. In contrast, we propose a single foundation model that generalizes across taxa. While recent self-supervised and audio-language contrastive models such as AVES (Hagiwara, 2023) and BioLingual (Robinson et al., 2024) have shown promising results on bioacoustics benchmarks, their discriminative and contrastive training paradigms constrain the range of tasks they can effectively address.

In recent years, foundation models—trained on large, diverse datasets, often via self-supervision—have shown promising performance across multiple domains (Bommasani et al., 2021). While transformer-based large language models (LLMs) are currently the most prominent examples, other architectures, such as diffusion models (Kingma et al., 2021), are also emerging as foundation models in some domains. Their ability to handle unseen tasks, perform in-context learning, and respond flexibly to prompts makes them as an appealing alternative to traditional machine learning methods, which typically depend on manually annotated datasets, extensive computational resources, and domain-specific expertise.

While multimodal LLMs, particularly vision-language models (VLMs), have been explored in biodiversity and conservation research (Miao et al., 2024), large audio-language models (LALMs) remain underexplored for bioacoustics. LALMs have demonstrated strong performance in human speech (Rubenstein et al., 2023; Wang et al., 2024; Wu et al., 2023a; Zhang et al., 2024), music (Gardner et al., 2024; Agostinelli et al., 2023), and general audio (Tang et al., 2024; Chu et al., 2024; Gong et al., 2024), and they hold significant potential for advancing bioacoustics as well.

In this paper, we introduce NatureLM-audio, the first audio-language foundation model specifically designed for bioacoustic tasks, including classification, detection, and captioning. Inspired by cross-taxa transfer observed in previous research, such as between human and gibbon or marmosets (Cauzinille et al., 2024; Sarkar & Magimai.-Doss, 2023) and birds and whales (Ghani et al., 2023), we incorporate speech and music tasks into training. We show that representations learned from these domains successfully transfer to animal vocalizations, demonstrating generalization across species. Additionally, we expand the BEANS bioacoustics benchmark (Hagiwara et al., 2023) with new tasks, including call-type prediction, lifestage classification, captioning, and individual counting. This new benchmark, BEANS-Zero, enables us to evaluate cross-domain learning and zero-shot transfer to unseen taxa and tasks. Unlike existing bioacoustics benchmarks such as Perch (Ghani et al. (2023) for bird detection) and BirdSet (Rauch et al. (2025) for bird classification), we do not focus solely on birds and we go beyond species classification. Additionally, our dataset presents prompts and audio descriptions in natural language, fostering further research in LALMs.

Our contributions are as follows: **(i) Model**: We introduce NatureLM-audio, the first audio-language foundation model for bioacoustics, trained on a carefully curated dataset spanning animal vocalizations, general audio, human speech, and music. **(ii) Domain transfer**: We show that our model generalizes beyond the species seen during training and exhibits zero-shot capabilities on unseen taxa and species. **(iii) Task transfer**: We evaluate our model on BEANS-Zero, which extends be-

yond species classification and includes unseen tasks such as individual counting. For the first time, we show positive transfer from speech and music data to bioacoustics tasks.

## 2    RELATED WORK

Most prior work on audio-language models has focused on human speech processing. Models such as SpeechGPT (Zhang et al., 2023), Speech-LLaMA (Wu et al., 2023a), AudioLM (Borsos et al., 2023), AudioPaLM (Rubenstein et al., 2023), AudioGPT (Huang et al., 2024b), SpiRit-LM (Nguyen et al., 2025), and SpeechLM (Zhang et al., 2024) mostly focus on building language models that can perceive and produce human speech. While such models could, in principle, be fine-tuned for bioacoustic tasks, doing so would require significant computational resources and domain expertise. Instead, our model shows promising generalization to unseen species and tasks without requiring additional fine-tuning.

Recently, more general language models with audio perception capabilities have emerged. Pengi (Deshmukh et al., 2023) uses an audio encoder and a text encoder mapped onto an LLM to perform audio-to-text tasks. SALMONN (Tang et al., 2024) uses dual audio encoders and integrates Q-Former (Li et al., 2023) to improve the handling of speech and general audio inputs. Qwen-audio (Chu et al., 2023) adopts a multi-task learning approach with the introduction of the Speech Recognition with Timestamp (SRWT) task. LTU (Gong et al., 2024) builds an open-ended question-answer dataset and applies curriculum learning strategies to improve generalization. Similar multimodal models have been proposed for music, such as MU-LLaMA (Liu et al., 2024) and LLark (Gardner et al., 2024). While recent foundation models such as AVES (Hagiwara, 2023) and BioLingual (Robinson et al., 2024) have demonstrated promising results in bioacoustics, their training paradigms and architectures constrain the range of tasks they can address.

Although animal sounds and vocalizations are often part of generic audio datasets such as AudioSet (Gemmeke et al., 2017) and audio caption datasets (Kim et al., 2019; Mei et al., 2024), these datasets are often too broad and lack the fine-grained annotations required for bioacoustic tasks such as species classification, behavior analysis, or ecological monitoring. As a result, LALMs trained on these datasets tend to produce only generic labels (e.g., 'bird' rather than a specific species name). We address this limitation by introducing a diverse, multi-task training dataset and NatureLM-audio, an LALM designed to produce robust representations for bioacoustics.

While specific bioacoustics benchmarks such as BIRB (Hamer et al., 2023) for bird vocalization retrieval and BEANS (Hagiwara et al., 2023) for classification and detection exist, the field still lacks comprehensive benchmarks comparable to those in human speech and music, such as Dynamic-SUPERB (Huang et al., 2024a) or AIR-Bench (Yang et al., 2024). This gap limits the evaluation of bioacoustic models, particularly in areas such as zero-shot learning and task generalization.

In this work, we aim to bridge these gaps by introducing NatureLM-audio, the first audio-language foundation model specifically designed for bioacoustics, and BEANS-Zero, an expanded benchmark that evaluates cross-species and cross-task generalization.

## 3    TRAINING DATASET CREATION

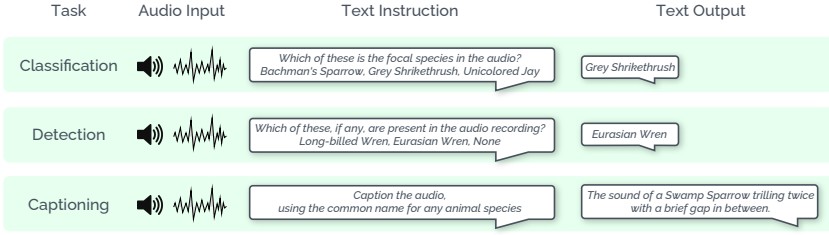

Figure 2: Examples of training instances.

| Task[a] | Dataset | # Hours | # Samples |
|---|---|---|---|
| CAP | WavCaps (Mei et al., 2024) | 7568 | 402k |
| CAP | AudioCaps (Kim et al., 2019) | 145 | 52k |
| CLS | NSynth (Engel et al., 2017) | 442 | 300k |
| CLS | LibriTTS (Zen et al., 2019) | 689 | 337k |
| | VCTK (Yamagishi et al., 2019) | | |
| CAP | Clotho (Drossos et al., 2020) | 25 | 4k |
| CLS, DET, CAP | Xeno-canto (Vellinga & Planqué, 2015) | 10416 | 607k |
| CLS, DET, CAP | iNaturalist (iNaturalist) | 1539 | 320k |
| CLS, DET, CAP | Watkins (Sayigh et al., 2016) | 27 | 15k |
| CLS, DET | ASA (Museum für Naturkunde Berlin) | 78 | 16k |
| DET | Sapsucker Woods (Kahl et al., 2022) | 285 | 342k |
| CLS, DET | Barkley Canyon (Kanes, 2021) | 876 | 309k |
| CLS | Urbansound (Salamon & Jacoby, 2014) | 10 | 2k |

Table 1: Training tasks and datasets. [a] CLS: classification, DET: detection, CAP: captioning.

To train an audio-text model for bioacoustics, we compile a diverse dataset of text-audio pairs (Table 1). The data is collected through a combination of prompting on existing audio datasets, generating new LLM-generated text labels, and mixing new, procedurally augmented audio data. The dataset is comprised of bioacoustic recordings, general audio, speech, and music datasets. Figure 2 shows examples of training instances used for NatureLM-audio. We plot the distribution of the training samples in Figure 3 in the Appendix.

## 3.1 BIOACOUSTIC DATA

**Species Classification**: We standardize large-scale bioacoustic archives into a common format, processing datasets such as Xeno-canto (Xeno-canto), iNaturalist (iNaturalist), Animal Sound Archive (Museum für Naturkunde Berlin), and Watkins (all-cuts, Sayigh et al. (2016)). Differences in species naming conventions across datasets are reconciled using the GBIF taxonomy backbone (GBIF Secretariat, 2023). We prompt the model to predict the scientific name, common name, or "taxonomic name" of the focal species or all species present in a recording. Taxonomic names are written as "phylum class order family genus species" and are inspired by BioCLIP (Stevens et al., 2024), which found that flattening the hierarchy into a text name improved generalization to unseen species in computer vision. In many real-world applications, an animal vocalization is known to belong to a subset of species—for example, based on geographic location. To model this, we generate prompts that present the model with a set of candidate species as possible answers. For 30% of prompts, we sample "random" negatives by selecting from all common names or scientific names in our dataset. In the remaining prompts, we introduce "hard" negatives by selecting species that share a common ancestor at the family, order, or phylum level. The number of negative samples is randomly selected, up to a maximum of 35.

Unlike traditional bioacoustic models that predict based on audio alone, the text-audio formulation enables classification conditioned on additional context. We train the model to classify species while conditioned on recording metadata and field notes. We follow the same setup as above, but inject the time of the recording, the location, and the free-text notes of the recordist into the prompt. This data is added wherever available for Xeno-canto, with time, location, or field note components randomly dropped a percentage of the time.

To avoid data leakage, we exclude a set of held-out species and the cbi data used in BEANS-Zero.

**Species Detection**: Using the same datasets as in species classification, we prompt the model to determine whether a given species is present in a recording. The model selects from a provided set of candidate species or chooses "None" when no correct option is given. Candidate sets are constructed with a mix of random and hard negatives, similar to the classification task. In 50% of prompts, the correct species is omitted from the set, making "None" the correct answer.

Because Xeno-canto comprises mostly focal recordings, we account for the covariate shifts in soundscapes by adding noise—audio that does not contain animal vocalizations, speech, or music. We source noise samples from datasets including: ShipsEar (Santos-Domínguez et al., 2016), Deepship (Irfan et al., 2021) and Orcalab (Poupard et al., 2020) for boat engine sounds, as well as

FSD50K (Fonseca et al., 2021) and Urbansound (Salamon & Jacoby, 2014) for non-animal, non-music sound classes, and all the classes from TUT2016 (Mesaros et al., 2016), IDMT (Abeßer et al., 2021), Demand (Thiemann et al., 2013), and Wham noise (Wichern et al., 2019). The noise is added programmatically, using random files at a random signal-to-noise ratio (SNR) sampled from a uniform distribution between $-10$dB and 20dB.

In addition, we used soundscape recording datasets for detection from Sapsucker Woods (SSW Kahl et al. (2022)) for birds and from Barkley Canyon (Kanes, 2021; Society, 2013; 2014a;b) for marine mammals. Following the BEANS detection dataset methodology, we segment audio into 10-second windows with a 5-second overlap, and treated it as a multi-label classification problem. Species with more than 100 occurrences were used as target labels, while those with fewer occurrences were grouped into an "other" class.

**Captioning**: For bioacoustic captioning, we use the AnimalSpeak dataset (Robinson et al., 2024), which aggregates bioacoustic datasets into a language-model-captioned dataset. We add separate prompts for captioning with scientific vs. common names, for "rich" captions over eight words, and for templated captions from Xeno-canto which follow a strict structure.

**Call-type and Lifestage**: We include multiple new bioacoustic tasks which can be expressed based on the Xeno-canto metadata. Specifically, predicting the life stage of birds, predicting call-types, and differentiating between calls and songs. The model is prompted using either audio alone or audio with the species name. Additionally, we include marine mammal call-type classification using Barkley Canyon recordings. These tasks go beyond species classification, providing finer-grained insights into ecological monitoring and animal behavior studies.

## 3.2 NON-BIOACOUSTIC DATA

**General Audio**   We include WavCaps (Mei et al., 2024), AudioCaps (Kim et al., 2019), and Clotho (Drossos et al., 2020) for general audio captioning. We observe that, during WavCaps creation, some recordings originally contained metadata relevant to bioacoustics and specific species. However, this metadata was lost during general-domain captioning, resulting in overly generic descriptions. We identify such cases by analyzing the original metadata, and re-process the metadata prompting Gemini-1.0-pro to produce bioacoustic captions. These enhanced captions are included alongside the original ones.

**Music**   Pitch, timbre, and the number of animals in a recording are key acoustic features used by biologists to infer context and behavior. We use NSynth 2.3.3 (Engel et al., 2017) to create a set of tasks that may help bioacoustics downstream tasks. We generate text prompts for *pitch detection* in Hz, *instrument name*, and *velocity*, ranging 0 to 1. Additionally, we use the timbre 'qualities' labels to create *text descriptions* for each audio. For instance, if the sound is 'distorted,' we generate descriptions such as "This sound has a distinctive crunchy sound and presence of many harmonics." or "This sound is distorted". Moreover, we create synthetic mixtures by layering one to three different instruments. In this case we generate two tasks: predicting the *number of instruments* and identifying the *instrument names*.

**Speech**   We use LibriTTS (Zen et al., 2019) and VCTK (Yamagishi et al., 2019) to generate synthetic mixtures of up to four speakers, a task that may transfer to individual counting in bioacoustics. To better match the frequency variability in animal vocalizations, we time-scale the speech mixtures with factors sampled from an uniform distribution between 0.25 to 4 (i.e., from 4x slower to 4x faster). Since animal vocalizations tend to be sparse, we insert random segments of silence at local minima computed on the RMS of the speech signals. To enhance realism, we further convolve the generated mixtures with impulse responses sampled from the DNS Challenge (Dubey et al., 2024b).

## 4   EVALUATION DATA: THE BEANS-ZERO BENCHMARK

One of the key contributions of this work is BEANS-Zero, a new benchmark for bioacoustics (Table 2). BEANS-Zero extends beyond traditional species classification by introducing new tasks such as call-type prediction, lifestage classification, captioning, and individual counting, which is not

| Task[a] | Dataset | Description | # Size[b] | # Labels (type) |
|---------|---------|-------------|-----------|------------------|
| CLS | esc50 | generic sound | 400 | 50 (sound type) |
| CLS | watkins | marine mammals | 339 | 31 (species) |
| CLS | cbi | birds | 3620 | 264 (species) |
| CLS | humbugdb | mosquito | 1859 | 14 (species) |
| DET | dcase | birds & mammals | 13688 | 20 (species) |
| DET | enabirds | birds | 4543 | 34 (species) |
| DET | hiceas | cetaceans | 1485 | 1 (species) |
| DET | rfcx | birds & frogs | 10406 | 24 (species) |
| DET | gibbons | gibbons | 18560 | 3 (call type) |
| CLS | unseen-species | birds etc. | 1255 | 200 (species) |
| CLS | unseen-genus | birds etc. | 951 | 101 (species) |
| CLS | unseen-family | birds etc. | 451 | 36 (species) |
| CLS | lifestage | birds | 466 | 3 (stage) |
| CLS | call-type | birds | 1000 | 2 (call/song) |
| CAP | captioning | birds etc. | 29002 | (open-ended) |
| CLS | zf-indv | zebra finches | 1160 | 2 (# of indv.) |

Table 2: Evaluation tasks and datasets of BEANS-Zero. [a] CLS: classification, DET: detection, CAP: captioning. [b] The numbers of samples for classification and captioning, and the number of 5-second "chunks" for detection (see Section 3 for more details.)

seen during training. To construct BEANS-Zero, begin with the test portion of BEANS (Hagiwara et al., 2023) evaluates models on standard bioacoustic tasks and datasets, including:

- esc50 (Piczak, 2015): Generic environmental sound classification with 50 labels.

- watkins (Sayigh et al., 2016): Marine mammal species classification with 31 species.

- cbi (Howard et al., 2020): Bird species classification with 264 labels from the Cornell Bird Identification competition hosted on Kaggle.

- humdubdb (Kiskin et al., 2021): Mosquito wingbeat sound classification into 14 species.

- dcase (Morfi et al., 2021): Mammal and bird detection from DCASE 2021 Task 5: Few-shot Bioacoustic Event Detection (20 species).

- enabirds (Chronister et al., 2021): Bird dawn chorus detection (34 species).

- hiceas (Center, 2022): Minke whale detection from the Hawaiian Islands Cetacean and Ecosystem Assessment Survey (HICEAS) (1 label).

- rfcx (LeBien et al., 2020): Bird and frog detection from the Rainforest Connection (RFCx) data with 24 species.

- gibbons (Dufourq et al., 2021): Hainan gibbon detection with 3 call type labels.

We also include novel bioacoustics datasets including:

- unseen-species: 200 species held out from AnimalSpeak (Robinson et al., 2024). For a controlled measure of generalization, we hold out species whose genus is well-represented (at least 150 training examples)

- unseen-genus: We hold out entire genus whose family is well-represented (at least 250 training examples) totaling 101 unique species.

- unseen-family: We hold out entire families whose class is well-represented (at least training 250 examples) totaling 36 unique species and representing the hardest generalization setting.

- lifestage: Predicting the lifestage of birds across multiple species. Newly curated from Xeno-canto (Xeno-canto).

- call-type: Classifying song vs. call across multiple bird species. Newly curated from Xeno-canto (Xeno-canto).

- captioning: Captioning bioacoustic audio on AnimalSpeak (Robinson et al., 2024).

- zf-indv (Elie & Theunissen, 2016): Determining whether a recording contains multiple zebra finches, using programmatically generated mixtures (1–4 individuals).

Some of these tasks, particularly bioacoustic captioning, have not been extensively studied before. Captioning allows for automatic generation of descriptive annotations of animal sounds, enhancing our understanding of species behaviors and communication patterns. Improvements in other new tasks, such as cross-species lifestage and call-type prediction, would allow finer-grained ecological monitoring and animal communication studies at scale.

For evaluation, we use accuracy for classification, macro-averaged F1 for detection, and SPIDEr (Liu et al., 2017) for captioning. Unlike mean average precision (mAP), which is originally used in BEANS and assumes a smooth ranking of candidates, F1 is more appropriate for evaluating generative tasks. This ensures a fairer assessment of models that generate predictions instead of ranking pre-defined classes.

## 5 NATURELM-AUDIO ARCHITECTURE

Our model follows a generic audio-to-text architecture similar to prior LALMs such as SALMONN (Tang et al., 2024), Qwen2-audio (Chu et al., 2024), and LTU (Gong et al., 2024). These models are trained on paired audio-text data for tasks including speech, music, and general audio event understanding. Figure 1 provides an overview of the NatureLM-audio architecture.

NatureLM-audio first encodes the input audio using BEATs (Chen et al., 2023), a state-of-the-art audio encoder on multiple audio tasks. To connect the BEATs embeddings with the LLM, we use a Q-Former (Li et al., 2023) applied at the window level as proposed in SALMONN (Tang et al., 2024). Similarly to the existing LALMs, we use an LLM to produce text, in this case Llama 3.1-8b (Dubey et al., 2024a), which is fine-tuned with LoRA (Hu et al., 2022). During training, only the adapter layers of the LLM are updated, while the base LLM parameters remain frozen. In contrast, the audio encoder and Q-Former remain trainable. The model takes an audio input $\boldsymbol{a}$ along with an instruction $\boldsymbol{x}$ and produces a text output $\boldsymbol{y}$. The model is trained under the loss function:

$$\boldsymbol{h} = f_W(\text{Encoder}(\boldsymbol{a})) \tag{1}$$

$$\boldsymbol{z} = p_\varphi^Q(\boldsymbol{q}, \boldsymbol{h}) \tag{2}$$

$$L = -\sum \log p_\theta^{LM}(\boldsymbol{y}_t | \boldsymbol{x}, \boldsymbol{z}, \boldsymbol{y}_{<t}) \tag{3}$$

where Encoder is the pretrained BEATs (Chen et al., 2023) audio encoder, $f_W$ is a function that converts consecutive $W$ audio frames into a window, $p_\varphi^Q$ is the Q-Former model with trainable parameters $\varphi$ that converts a window into a sequence of text representations $\boldsymbol{z}$ using query $\boldsymbol{q}$, and $p_\theta^{LM}$ is the pretrained LLM with trainable parameters $\theta$.

## 6 TRAINING METHOD

Our training method follows a curriculum learning approach (Soviany et al., 2021), where the model is first trained on simpler tasks before progressively tackling more complex ones, as done in other audio foundation models (Tang et al., 2024; Gong et al., 2024). We train in the two stages:

- Stage 1 (Perception Pretraining): We pretrain the model exclusively on focal species classification, classifying vocalizations from thousands of animal species. Species classification is a highly deterministic task, allowing opportunity to learn a robust connection between language and audio. We also choose to train on this task individually as it is foundational to other tasks in bioacoustics.

- Stage 2 (Generalization Fine-tuning): In the second stage, we introduce a variety of bioacoustic and other tasks, building on the robust classification abilities developedin Stage 1. This includes detection, captioning, lifestage prediction, and call-type prediction. We also include speech and music data in this second stage, aimed at improving transfer to bioacoustic tasks.

We train NatureLM-audio from scratch, initializing the Q-Former and LoRA layers randomly rather than fine-tuning existing LALM checkpoints such as SALMONN. This allows for more flexibility in terms of choosing the latest LLM with the extensive knowledge of animal species, and the most relevant architectural components (e.g., excluding memory-intensive parts of current LALMs such as the Whisper speech encoder (Radford et al., 2023)).

| Model | esc50 | watkins | cbi | humbugdb | dcase | enabirds | hiceas | rfcx | gibbons |
|---|---|---|---|---|---|---|---|---|---|
| LLM w/o audio | 0.020 | 0.041 | 0.005 | 0.073 | 0.000 | 0.001 | 0.210 | 0.000 | 0.013 |
| SALMONN | 0.320 | 0.041 | 0.004 | 0.090 | 0.005 | 0.004 | 0.097 | 0.002 | 0.005 |
| Qwen2-audio | 0.307 | 0.041 | 0.004 | 0.070 | 0.005 | 0.004 | 0.097 | 0.002 | 0.005 |
| BioLingual | 0.600 | 0.257 | 0.705 | 0.085 | 0.036 | 0.109 | **0.429** | 0.004 | **0.018** |
| NatureLM-audio | **0.820** | **0.788** | **0.778** | **0.114** | **0.058** | **0.314** | 0.336 | **0.025** | 0.005 |

Table 3: Main zero-shot results on BEANS-Zero. We used accuracy for classification, and F1 for detection tasks. The best and the second best metrics are highlighted and underlined per each dataset.

# 7 EXPERIMENTS

## 7.1 TRAINING AND EVALUATION DETAILS

We train our model on the full curated training set (Section 3). To evaluate generalization, we create hold-out splits for Xeno-canto, iNaturalist, Animal Sound Archive, and Watkins datasets, used solely for benchmarking.

We initialize the audio encoder weights using an existing BEATs checkpoint (`BEATs_iter3_plus_AS2M_finetuned_on_AS2M_cpt2.pt`) and fully fine-tune it, which we found to be critical in an ablation (Table 9). We initialize the LLM from Llama-3.1-8B-Instruct and apply LoRA to all attention layers (rank: 32, alpha: 32, dropout: 0.1).

We follow the proposed two-stage training strategy. In both stages, we use a linear warmup followed by a cosine learning rate schedule, with a peak learning rate of $9.0 \times 10^{-5}$ and an end learning rate of $2.0 \times 10^{-5}$. We use a batch size of $128$ and run the first stage for $5.0 \times 10^{5}$ steps and the second stage for $1.6 \times 10^{6}$ steps. For inference, we use beam search with two beams, a repetition penalty of 1.0, and a length penalty of 1.0.

We consider several inference methods depending on the task type. Species-classification tasks involve single-label prediction: we prompt the model to output the species name from the recording. Since the LLM may generate text that does not exactly match predefined labels, we use Levenshtein distance to map predictions to the closest species name. We choose the Levenshtein distance for its simplicity and because species names, in particular Latin names, have high character-overlap with related names. However, we note that it may not be optimal for general audio classification.

For multilabel detection tasks, the number of target species varies by dataset. For tasks with 10 or fewer species, we include the species options in the prompt. Otherwise we prompt the model to list all species in the audio, if any. In both cases, the model outputs all detected species, or 'None'. We discard predictions with low character overlap with the valid labels.

Our baselines include CLAP-like models (Wu et al., 2023b), which cannot naively perform multilabel detection. To address this, we create a negative "template" for each detection task, as proposed by Miao et al. (2023). We consider each label a detection positive for CLAP if the audio is more similar to the label than to the negative template in the CLAP model's embedding space.

## 7.2 SPECIES CLASSIFICATION AND DETECTION

Table 3 shows the main results measured on the BEANS-Zero species classification and detection datasets. Our baselines include an LLM (the original Llama-3.1-8B-Instruct model without fine-tuning, Dubey et al. (2024a)) without audio input, SALMONN (Tang et al., 2024), BioLingual (Robinson et al., 2024), and Qwen2-audio (Chu et al., 2024). All baselines are evaluated in the same way as NatureLM-audio. As shown in the table, the outputs from the LLM without audio input, SALMONN, and Qwen2-audio are largely random on bioacoustic datasets, failing to properly interpret the input audio or follow the instructions. In contrast, NatureLM-audio achieved state-of-the-art zero-shot performance on 7 out of 9 datasets, and delivered competitive results on the remaining tasks from the BEANS-Zero benchmark. We note that performance of baselines on the general audio dataset ESC50 (Piczak, 2015) may be reduced by the use of the Levenshtein distance, as our pipeline is optimized for bioacoustic tasks.

|  | cbi | dcase-bird | enabirds |
|---|---|---|---|
| BirdNET | 0.609 | 0.035 | 0.490 |
| Perch | 0.744 | 0.035 | 0.164 |
| NatureLM-audio | 0.778 | 0.083 | 0.314 |

Table 4: Comparison with bird vocalization models.

|  | unseen-species[a] | unseen-genus[b] | unseen-family[c] |
|---|---|---|---|
| Supervised SotA | 0.687 | 0.688 | 0.545 |
| random chance | 0.005 | 0.010 | 0.028 |
| baseline (CLAP) | 0.014 | 0.026 | 0.082 |
| NatureLM-audio (cmn) | 0.181 | 0.116 | 0.035 |
| NatureLM-audio (sci) | 0.238 | 0.041 | 0.035 |
| NatureLM-audio (tax) | **0.343** | **0.148** | **0.308** |

Table 5: Generalization to unseen taxa in terms of classification accuracy. All tasks predict species names, on test sets held-out at the [a] species [b] genus and [c] family level. Targets were not held out from "Supervised SotA" reference (BioLingual). Cmn, sci, and tax denote predictions using common, scientific, and taxonomic names respectively. Since the number of labels varies across datasets, results should not be directly compared across columns.

We also compared NatureLM-audio with bird-specific classification models, namely BirdNET (Kahl et al., 2021) and Perch (Ghani et al., 2023), to evaluate the zero-shot capabilities of our model. We compare on the bird-related datasets of BEANS-Zero, plus the portion of DCASE with bird species. Results are presented in Table 4. Since both BirdNET and Perch were trained in a supervised manner on datasets that significantly overlap with our bird evaluation datasets, this is not a fully fair comparison, and their performance should be considered as topline results. Nevertheless, our model demonstrated strong zero-shot bird vocalization classification capabilities. In particular, we achieve a new SotA for the cbi dataset, classifying vocalizations of hundreds of birds, and achieve competitive results with the bird-specific models on both detection tasks. We additionally compare against various models on datasets from the BirdSet benchmark (Rauch et al. (2025), where our model achieves the highest average top-1 accuracy (Appendix in Table A.4).

## 7.3 GENERALIZING TO UNSEEN SPECIES

We further evaluate the model's ability to generalize to completely unseen taxa using the newly added datasets in BEANS-Zero, held out at three levels: unseen species, unseen genus, and unseen families. As a topline, we compare against BioLingual, which has seen these taxa in training and only indicates fully supervised performance. As baselines, we consider a theoretical random baseline (1 / number of classes) and CLAP-LAION (Elizalde et al., 2023), a general-domain audio model which, similar to our model, is unlikely to have seen these species during training. We compare the performance when predicting common, scientific, or taxonomic names.

Table 5 presents the results. Across all three unseen taxa settings, NatureLM-audio significantly outperforms the random baseline, demonstrating its ability to generalize to unseen taxa and taxonomic branches. For example, on the unseen species test set, our model achieves an accuracy of 34.3%, far surpassing the random baseline of 0.5%, indicating that the model has learned features that extend beyond the species it was trained on. The model also outperforms CLAP-LAION, further emphasizing its ability to generalize. We observe that predicting with taxonomic names consistently improves performance across all settings, and is particularly critical for generalizing to unseen genus and families where scientific (Latin) names alone fail to capture hierarchical relationships. We further note that scientific names perform relatively well when generalizing to unseen species, but perform worse than common names for generalizing to unseen genus, This suggests that common names may encode broader hierarchical information or be more familiar to the language model.

## 7.4 NOVEL BIOACOUSTIC TASKS

Beyond species classification, we evaluate NatureLM-audio on novel bioacoustic tasks introduced in BEANS-Zero, which, to the best of our knowledge, have not been previously studied at a cross-

|  | lifestage | call-type | captioning | zf-indv |
|---|---|---|---|---|
| SotA | 0.502 | 0.658 | 0.009 | 0.604 |
| NatureLM-audio | 0.794 | 0.871 | 0.532 | 0.655 |

Table 6: Results on BEANS-Zero novel bioacoustics tasks. We report accuracy for classification, and SPIDEr (Sharif et al., 2018) for captioning. SotA is SALMONN for captioning and BioLingual for the remaining tasks.

species level. We additionally include `zf-indv`, a completely unseen task that determines whether a recording contains multiple zebra finch individuals or just one (Elie & Theunissen, 2016). We compare against BioLingual (Robinson et al., 2024) for discriminative tasks and SALMONN (Tang et al., 2024) for captioning. As shown in Table 6, NatureLM-audio sets a new state-of-the-art across all tasks. We evaluate call-type classification more extensively (Table 8), and find the model is able to transfer this task to unseen taxa. We further find the model can improve audio classification performance by incorporating additional context as text, which we discuss in the Appendix in A.5.

## 7.5 ABLATION ON SPEECH AND MUSIC

To investigate the impact of speech and music on downstream task performance, we run an ablation during stage-2 training. Specifically, we train two versions of the model for 150k steps—one with speech and music data and one without—and evaluate their ability to perform an unseen task: counting zebra finches. The model trained with speech achieves 67.7%, similar to our full model. The model trained without speech scored 50.0%, exactly random, and qualitatively predicted 'more than one' for all examples. These results suggest the ability to count vocalizing birds transfers from human speech and music, as our training data includes tasks such as counting human speakers in a recording. We include the ablation performance on all tasks in the Appendix (Tables 10 and 11).

## 8 CONCLUSION

We presented NatureLM-audio, the first audio-language foundation model specifically designed for bioacoustics, demonstrating its potential to address critical tasks such as classifying and detecting animal vocalizations, and decoding context, call types, and individuals across species. By leveraging a carefully curated dataset spanning bioacoustics, speech, and music data, NatureLM-audio sets the new state-of-the-art on multiple tasks, including zero-shot classification of unseen species. Moreover, our model demonstrates positive transfer across both domains and tasks, performing well on a novel benchmark (BEANS-Zero), which includes new bioacoustic tasks such as captioning and individual counting. To further accelerate research and the development of more robust models in the field, we have open-sourced the code for generating both training and benchmarking data.

We plan to extend this work by incorporating more diverse tasks and datasets, improving the text-based LLM backbone with bioacoustic-specific texts, and enhancing the model's multilingual capabilities. Another direction is the introduction of new modalities, such as motion and image data, leading to multimodal models like NatureLM-motion and NatureLM-image. We also aim to explore the model's generative abilities, particularly in producing audio tokens for applications such as animal sound synthesis and audio denoising.

While NatureLM-audio offers significant potential for advancing biodiversity monitoring and conservation, several ethical concerns must be addressed. First, there is a potential bias towards bird vocalizations due to the overrepresentation of bird datasets, which could limit the model's effectiveness in other taxa. Second, the model's ability to detect and classify endangered species could be misused for illegal activities such as poaching, posing a threat to wildlife. Finally, unintended consequences on animal behavior and ecology must be considered, particularly when deploying LLMs, known for their issues including hallucinations and biases (Kuan et al., 2024). These systems may interfere with the behavior of the species being studied, and the long-term ecological impact of widespread passive monitoring is still unknown. Careful deployment and responsible use are essential to mitigate these risks.

**Acknowledgement:** We thank Benno Weck, Sara Keen, Milad Alizadeh, Gagan Narula, and Matthieu Geist for their contributions to this work.

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

# A APPENDIX

## A.1 HELD-OUT FAMILIES

1. Elachuridae
2. Calyptophilidae
3. Pelecanoididae
4. Phocoenidae
5. Alytidae
6. Castoridae
7. Dicroglossidae
8. Suidae
9. Prophalangopsidae
10. Octodontidae

## A.2 HELD-OUT GENUS

1. Aglaeactis
2. Drepanorhynchus
3. Lesbia
4. Nemobius
5. Meconema
6. Pseudochorthippus
7. Caliechthrus
8. Pachycare
9. Rhodothraupis
10. Astrapia
11. Probosciger
12. Amazonetta
13. Ocyalus
14. Nandayus
15. Rhinocrypta
16. Heterocercus
17. Jacamaralcyon
18. Hymenops
19. Doliornis
20. Eugerygone
21. Cryptosylvicola
22. Taeniopygia
23. Catharopeza
24. Eurostopodus
25. Tylas
26. Vini
27. Ptychoramphus
28. Speculanas
29. Aphelocephala
30. Stipiturus
31. Procarduelis
32. Rhopophilus
33. Neopsephotus
34. Enodes
35. Leucocarbo
36. Gymnophaps
37. Goldmania
38. Oreomystis
39. Rhodostethia
40. Falcipennis
41. Pachycoccyx
42. Cryptotympana
43. Tympanistalna
44. Cyrtoxipha
45. Afrixalus
46. Uperoleia
47. Urocitellus
48. Chalcorana
49. Aiolopus
50. Speothos

## A.3 Held-out Species

1. Aethopyga shelleyi
2. Arachnothera dilutior
3. Sitta castanea
4. Carpodacus rodopeplus
5. Aethopyga ignicauda
6. Pachycephala soror
7. Herpsilochmus roraimae
8. Amazona dufresniana
9. Metallura aeneocauda
10. Thlypopsis fulviceps
11. Monarcha frater
12. Kleinothraupis reyi
13. Aplonis magna
14. Phylloscopus misoriensis
15. Agapornis pullarius
16. Amazona versicolor
17. Saltator cinctus
18. Xiphocolaptes falcirostris
19. Passer insularis
20. Chalcomitra balfouri
21. Arremonops tocuyensis
22. Atlapetes meridae
23. Colluricincla obscura
24. Saltator maxillosus
25. Philemon meyeri
26. Thamnophilus insignis
27. Aulacorhynchus whitelianus
28. Sirystes subcanescens
29. Sporophila nigrorufa
30. Zoothera mollissima
31. Thlypopsis inornata
32. Picumnus spilogaster
33. Columba arquatrix
34. Petrochelidon rufocollaris
35. Pyrrhura griseipectus
36. Myiothlypis chrysogaster
37. Thripophaga amacurensis
38. Herpsilochmus motacilloides
39. Progne dominicensis
40. Heliodoxa branickii
41. Asthenes arequipae
42. Gerygone fusca
43. Otus thilohoffmanni
44. Inezia subflava
45. Charadrius montanus
46. Petroica polymorpha
47. Symposiachrus vidua
48. Dicrurus lophorinus
49. Pycnonotus penicillatus
50. Melanerpes herminieri
51. Zosterops mysorensis
52. Oenanthe xanthoprymna
53. Artamus monachus
54. Caprimulgus pulchellus
55. Psarocolius cassini
56. Symposiachrus infelix
57. Zosterops cinereus
58. Circus cinereus
59. Geotrygon chrysia
60. Microspingus trifasciatus
61. Pternistis harwoodi
62. Ceblepyris caesius
63. Ficedula disposita
64. Treron affinis
65. Geokichla wardii
66. Campethera bennettii
67. Alcedo semitorquata
68. Buteo japonicus
69. Apus bradfieldi
70. Pterocles personatus
71. Melaniparus fringillinus
72. Poecile hypermelaenus
73. Circus buffoni
74. Pycnonotus blanfordi
75. Machlolophus aplonotus
76. Estrilda ochrogaster
77. Touit batavicus
78. Mirafra gilletti
79. Pternistis icterorhynchus
80. Accipiter collaris
81. Knipolegus lophotes
82. Nothoprocta taczanowskii
83. Pachycephala modesta
84. Vanellus tricolor
85. Caprimulgus andamanicus
86. Ardenna grisea
87. Mixornis kelleyi
88. Cinnyris johannae
89. Recurvirostra novaehollandiae
90. Sitta leucopsis
91. Petroica pusilla
92. Amazilia luciae
93. Melaniparus fasciiventer
94. Egretta picata
95. Columba pollenii
96. Rallus madagascariensis
97. Heliodoxa gularis
98. Carpodacus roseus
99. Zosterops chloronothos
100. Pachycephala lorentzi
101. Saucerottia cyanura
102. Cinclosoma marginatum
103. Bucco noanamae
104. Certhia nipalensis
105. Pachycephala lanioides
106. Carpodacus trifasciatus
107. Chorthippus acroleucus
108. Chlidonias albostriatus
109. Hirundo domicola
110. Falco concolor
111. Dryocopus schulzii
112. Rhyticeros undulatus
113. Quiscalus nicaraguensis
114. Cisticola brunnescens
115. Knipolegus cyanirostris
116. Ardenna carneipes
117. Lybius rubrifacies
118. Climacteris melanurus
119. Puffinus opisthomelas
120. Manorina melanotis
121. Celebesica abbotti
122. Otus mayottensis
123. Trachyphonus margaritatus
124. Oenanthe dubia
125. Chloropsis flavipennis
126. Ploceus alienus
127. Phalacrocorax varius
128. Ploceus pelzelni
129. Merops mentalis
130. Passer gongonensis
131. Myzomela cineracea
132. Pachycephala feminina
133. Brachypteryx sinensis
134. Lonchura flaviprymna
135. Ninox natalis
136. Myrmelastes caurensis
137. Buteo trizonatus
138. Apalis chariessa
139. Ficedula nigrorufa
140. Pica mauritanica
141. Anthreptes reichenowi
142. Sholicola major
143. Vireo osburni
144. Anas capensis
145. Ducula luctuosa
146. Lanius newtoni
147. Odontophorus dialeucos
148. Bostrychia olivacea
149. Cinnyris tsavoensis
150. Ploceus heuglini
151. Myzomela nigrita
152. Falco cherrug
153. Ixobrychus sturmii
154. Rhipidura semirubra
155. Haematopus chathamensis
156. Anthus brachyurus
157. Oenanthe lugens
158. Columba rupestris
159. Rhyticeros subruficollis
160. Zosterops vellalavella
161. Anthus sokokensis
162. Phaethornis idaliae
163. Picus dedemi
164. Muscicapa segregata
165. Cyanomitra bannermani
166. Polioptila facilis
167. Platysteira albifrons
168. Dicaeum pygmaeum
169. Puffinus assimilis
170. Rhipidura kubaryi
171. Ploceus katangae
172. Canis lupaster
173. Hyla andersonii
174. Ranoidea nudidigita
175. Ranoidea aurea
176. Litoria tyleri
177. Dendropsophus joannae
178. Okanagana occidentalis
179. Litoria latopalmata
180. Magicicada tredecassini
181. Orchelimum silvaticum
182. Oecanthus celerinictus
183. Empidonomus aurantioatrocristatus
184. Bufotes boulengeri
185. Oecanthus nigricornis
186. Myrmothera fulviventris
187. Psaltoda adonis
188. Rana dalmatina
189. Dendropsophus sanborni
190. Hyperolius stictus
191. Hyperolius pictus
192. Hyla eximia
193. Leptodactylus natalensis
194. Oecanthus californicus
195. Hyperolius parallelus
196. Gryllus cohni
197. Physeter macrocephalus
198. Eleutherodactylus unicolor
199. Gryllus bermudensis
200. Anas penelope

## A.4 EVALUATION ON BIRDSET

We evaluate Top-1 accuracy on the datasets from the BirdSet benchmark. To match other models evaluated on BirdSet, which are constrained to predict one of the allowed labels, we use loss-based classification across all datasets and make predictions using scientific names. Our model achieves the highest average Top-1 accuracy, slightly surpassing Perch, demonstrating strong generalization from primarily focal recordings to soundscape recordings, and state-of-the-art performance for retrieval and classification on real-world bird datasets.

| | POW | PER | NES | UHH | HSN | NBP | SNE | AVG |
|---|---|---|---|---|---|---|---|---|
| EffNet | 0.80 | 0.38 | 0.49 | 0.42 | **0.59** | 0.63 | 0.67 | 0.57 |
| ConvNext | 0.75 | 0.36 | 0.45 | 0.44 | 0.52 | 0.64 | 0.65 | 0.54 |
| AST | 0.79 | 0.40 | 0.48 | 0.39 | 0.48 | 0.61 | 0.57 | 0.53 |
| EAT | 0.69 | 0.32 | 0.46 | 0.40 | 0.47 | 0.61 | 0.58 | 0.50 |
| W2V2 | 0.72 | 0.34 | 0.47 | 0.51 | 0.50 | 0.65 | 0.51 | 0.53 |
| Perch | 0.85 | 0.48 | **0.66** | 0.57 | 0.58 | **0.69** | 0.69 | 0.65 |
| NatureLM-audio | **0.95** | **0.62** | 0.47 | **0.60** | 0.58 | 0.66 | **0.76** | **0.66** |

Table 7: Top-1 Accuracy results for each method on the datasets of BirdSet. Refer to the original paper (Rauch et al., 2025) for the details of compared baseline models.

## A.5 SPECIES CLASSIFICATION WITH ADDITIONAL CONTEXT

We evaluate whether NatureLM-audio can improve species classification performance by incorporating additional context as text. The CBI dataset (Howard et al., 2020), derived from Xeno-canto, often contains metadata such as location and free-text notes written by recordists. We evaluate the model under three conditions: using audio alone, adding metadata (latitude, longitude, altitude when available, and geographic region), and further incorporating free-text notes. The model achieves an accuracy of 0.776 with audio alone, 0.792 with additional metadata, and 0.798 with both metadata and free-text notes, demonstrating that providing additional textual context can improve audio classification performance.

## A.6 CALL TYPES AND TRANSFER

| Configuration | call-song | multi | call-song-unseen | multi-unseen |
|---|---|---|---|---|
| NatureLM-audio | **0.871** | **0.667** | **0.769** | **0.678** |
| BioLingual | 0.658 | 0.303 | 0.665 | 0.419 |

Table 8: Accuracy of call vs. song classification (call-song), multi-call classification (multi), and the generalization of these tasks to unseen taxa (call-song-unseen, multi-unseen.)

We further evaluate the classification of bird call types and the transfer of this task across species. We test the model on call vs. song prediction as well as call-type prediction for multiple classes (call, song, flight call, alarm call, begging call, and drumming). We then test if these tasks can be transferred to unseen taxa. The call-song-unseen and multi-unseen datasets evaluate the same tasks described above, but evaluated on the held-out taxa used to test unseen species, unseen genus, and unseen family. In addition to achieving state-of-the-art results on these tasks, the results transfer strongly to unseen taxa, outperforming BioLingual—even when these taxa were held out from NatureLM-audio but not from BioLingual.

## A.7 ABLATION ON UNFREEZING BEATs

| Configuration | watkins | cbi | unseen-species | unseen-genus | unseen-family |
|---|---|---|---|---|---|
| BEATs-unfrozen | **0.723** | **0.680** | **0.320** | **0.124** | **0.306** |
| BEATs-frozen | 0.490 | 0.401 | 0.184 | 0.073 | 0.186 |

Table 9: Zero-shot classification results with BEATs unfrozen vs. frozen. Both models are trained on stage-1 tasks for 150k steps. We report accuracy on species classification tasks, with unseen taxa tasks predicted using taxonomic names.

## A.8 SPEECH+MUSIC ABLATION: FULL RESULTS

| Model | esc50 | watkins | cbi | humbugdb | dcase | enabirds | hiceas | rfcx | gibbons |
|---|---|---|---|---|---|---|---|---|---|
| base | 0.570 | 0.788 | 0.748 | 0.093 | 0.107 | 0.299 | 0.415 | 0.038 | 0.011 |
| base w/o speech or music | 0.605 | 0.773 | 0.750 | 0.152 | 0.040 | 0.293 | 0.417 | 0.038 | 0.012 |

Table 10: Zero-shot classification and detection results on BEANS-Zero. Base model was trained on all stage-2 training tasks, while "base w/o speech or music" is an ablation removing both speech and music tasks from training data. Both models were trained for 150k steps. We used accuracy for classification, and F1 for detection tasks.

| Model | unseen-species | unseen-genus | unseen-family | lifestage | call-type | captioning | zf-indv |
|---|---|---|---|---|---|---|---|
| base | 0.322 | 0.139 | 0.239 | 0.702 | 0.863 | 0.501 | 0.677 |
| base w/o speech or music | 0.354 | 0.137 | 0.330 | 0.690 | 0.852 | 0.503 | 0.500 |

Table 11: Zero-shot results on new tasks introduced in BEANS-Zero. Base model was trained on all stage-2 training tasks, while base w/o speech or music is an ablation removing both speech and music tasks from training data. Both models were trained for 150k steps. We report accuracy for classification, and SPIDEr (Sharif et al., 2018) for captioning.

## A.9 TRAINING DATA

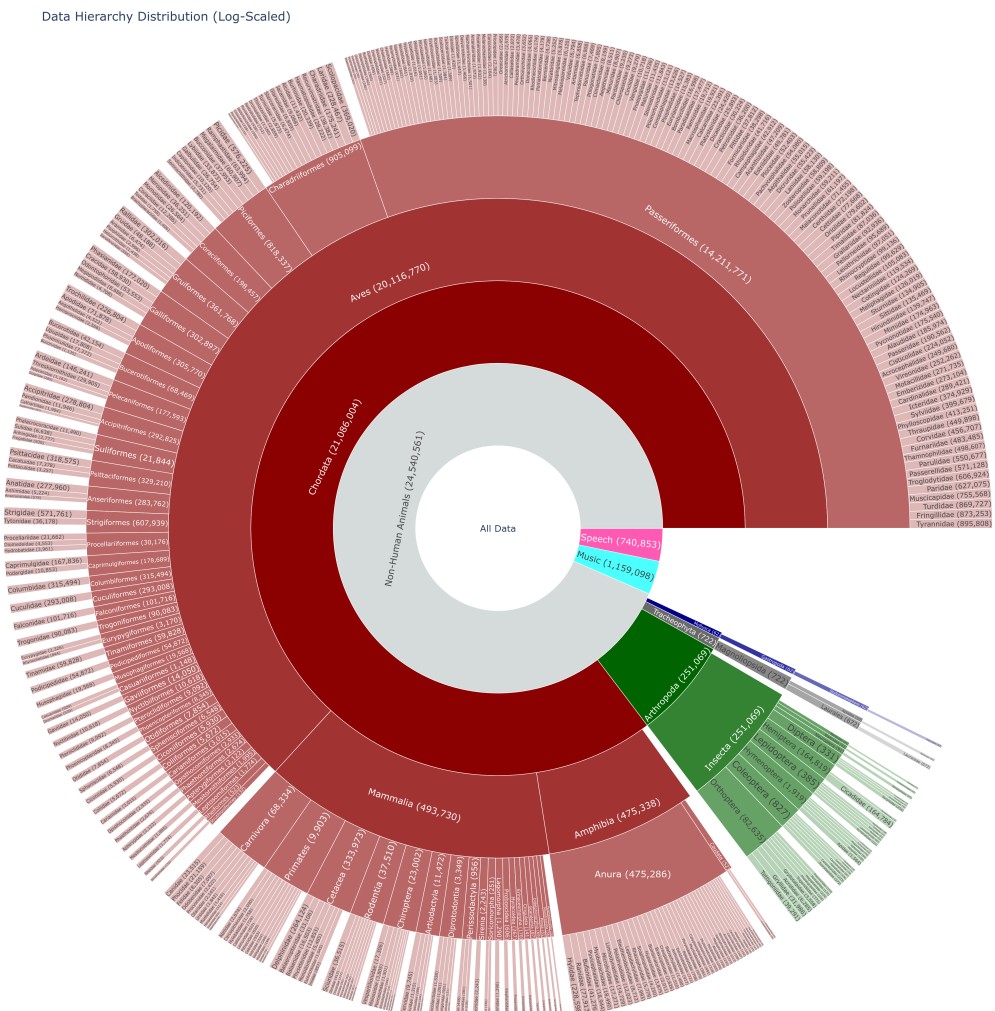

Figure 3: Data composition across training samples including the distribution for the main data types and phylum, class, and order for non-human animals. The counts represent prompts rather than audio files i.e. various prompts may be derived from the a single audio file.

