# OpenReview forum: "NatureLM-audio: an Audio-Language Foundation Model for Bioacoustics"
_ICLR.cc/2025/Conference — ICLR 2025 Poster_

### Official Review · Reviewer_1ReQ · 2024-10-21

**Soundness:** 2
**Presentation:** 3
**Contribution:** 2
**Rating:** 3
**Confidence:** 5

**Summary:**

The paper introduces NatureLM, an audio-language model specifically designed for bioacoustics, focusing on animal sounds. NatureLM leverages a diverse dataset combining bioacoustics, human speech, and music to enhance generalization across various species and tasks. NatureLM demonstrates strong in-context learning capabilities, enabling zero-shot classification for unseen species.

Additionally, the paper presents BEANS-Zero, a benchmark for bioacoustics that includes several tasks beyond species classification, such as call-type prediction, life-stage classification, and individual counting, aiming to push the boundaries of bioacoustic research.

**Strengths:**

1. The introduction of NatureLM, the first audio-language model specifically designed for bioacoustics, represents a promising new direction for incorporating language models into biodiversity monitoring.

2. The development of the BEANS-Zero benchmark extends the original BEANS benchmark by introducing new tasks, such as call-type prediction, life-stage classification, individual counting, and open-ended audio captioning. These additions have the potential to advance bioacoustics research and enable more detailed acoustic monitoring of species.

3. The paper is easy to follow, and the related works are thoroughly referenced.

**Weaknesses:**

1. Incorrect Terminology


1.1.  The introduction describes BioLingual as self-supervised; however, the supervision is derived from text generated based on class labels. I recommend referring to it as supervised learning with language-based supervision for greater clarity and accuracy.

2.1. Both BioLingual and AVES are described in the paper as foundation models, but this classification may be misleading. BioLingual and AVES are trained on datasets with less than 2 million samples, while models trained on AudioSet with 2 million samples are not typically considered foundation models. BioLingual is evaluated on classification and retrieval tasks, while AVES is evaluated on classification and detection tasks. Typically, a foundation model is a very large model trained on extremely large datasets so that properties emerge that enable it to handle a wide variety of tasks across different domains. For improved clarity, I suggest either using alternative descriptions like self-supervised (for AVES) and audio-language contrastive (for BioLingual) to more accurately describe them, or providing a clear definition of what you consider to be a foundation model if you wish to retain this terminology. This will help avoid confusion and ensure that readers more precisely understand the capabilities of these models.

2. Overstatement of Results


2.1. The analysis of NatureLM-audio's performance on the cbi dataset (Table 4, Section 4.2) is potentially misleading due to data overlap. Since BirdNet, Perch, and NatureLM-audio are trained on Xeno-Canto, and the cbi evaluation dataset is a subset of Xeno-Canto, there is overlap between the training and evaluation data. This compromises the claim of state-of-the-art performance, especially that Perch significantly outperforms NatureLM in enabirds.

2.2. In Table 5, Section 4.3, the authors claim that NatureLM-audio demonstrates generalization to completely unseen species by outperforming CLAP, a contrastive audio-language model trained on non-bioacoustic data. However, this conclusion seems too strong. The results primarily show that a model trained on bioacoustic data performs better than one trained on non-bioacoustic data, which does not necessarily indicate true generalization to unseen species. Additionally, BioLingual, which was trained similarly to CLAP but on bioacoustic data, significantly outperforms NatureLM-audio. I suggest rephrasing the conclusion to more accurately reflect these findings.

2.3. In Section 4.4, the paper claims state-of-the-art performance in the captioning task. However, this comparison is based solely on the SALMONN model, which was not trained on bioacoustic data and is not specifically designed for this task. To support the claim of state-of-the-art performance, I recommend including a comparison with a captioning model trained on bioacoustic data, as this would provide a more accurate and meaningful evaluation.

**Questions:**

1. In Section 3.1.1, the method for hard negatives sampling for species detection is briefly mentioned, but the details are unclear. Could you provide a more detailed explanation of the strategy for sampling these hard negative samples?

2. Could you provide more details in Section 3.2 about the selection strategy for the held-out species in the AnimalSpeak dataset for the unseen-cmn and unseen-sci subsets? Even if the selection was random, it's important to clarify the process.

3. In Section 4.5, you state that including speech and music in training has a positive impact on counting zebra-finch individuals. However, while the ablation study includes speech, it is unclear whether music was also ablated. Could you clarify whether the effect of music was tested? Additionally, conducting ablation on all downstream tasks, not just individual identification, could provide more comprehensive insights into whether speech and music data enhance performance across other tasks as well. This could help clarify which types of training data are most beneficial for specific applications.

---

> ### Author Response · Authors · 2024-11-21
>
> We thank the reviewer 1ReQ for the comments which help improve the manuscript. We appreciate that the reviewer found the document easy to read, and identified our core contributions: training and benchmarking data, in addition to the first bioacoustics LALM.
>
> Next, we respond to each of the issues raised.
>
> Weaknesses:
>
> > 1. Incorrect terminology
>
> > 1.1 Self-supervision in Biolingual
>
> We agree that we should clarify that BioLingual is trained via contrastive learning. We’ve made this change in the paper.
>
> > 1.2 Biolingual and AVES are not foundation models
>
> We have rephrased the paper using the alternative descriptions suggested by the reviewer. Note that we use the foundation model definition from the following paper which we cited in the corresponding paragraph: [Bommasani, et al. 2021](https://arxiv.org/abs/2108.07258).
>
> > Overstatement of results
>
> > 2.1 Comparison to BirdNET and Perch, cbi-XC leakage
>
> We acknowledge the concern regarding data overlap between training and evaluation datasets. As clarified in Section 3.1.1, we have excluded the CBI test set from our training data, ensuring that our results are valid and free from data leakage. Additionally, we apologize for the error mistakenly switching the baseline results of Perch and BirdNet on the enabirds dataset; this has been corrected in the paper. The corrected results show that NatureLM-audio performs best on two of the three datasets (CBI and DCASE-bird), while BirdNet achieves highest performance on enabirds. However, BirdNet training data are not readily available online, and [Chasman et al. (2024)](https://openreview.net/forum?id=QCY01LvyKm) identified BirdNet had likely been trained on Powdermill, which overlaps with enabirds. This could explain BirdNet’s superior performance on this dataset, as there is potential leakage. For this reason, we view BirdNet and Perch as specialized toplines rather than baselines.
>
> > 2.2 Generalization to unseen species
>
> Generalizing to unseen species would mean better-than-random performance on our test set of species held out from training. Using scientific names, our model achieves 19.6% which is well above random performance of 1/300 or ~0.3% on this dataset, sufficient to conclude the model is able to generalize to unseen species. Importantly, BioLingual was trained on these species and results cannot be compared, we include it only as a reference for fully-supervised performance. We have updated the text to include the random baseline and make the comparisons and conclusions more clear.
>
> > 2.3 Captioning baseline
>
> To our best knowledge, there is no existing captioning model trained for bioacoustics and this is one of the features of NatureLM-audio. We believe that training a new captioning baseline for these data is the scope of a new paper, rather than a separate experiment. We did add the comparison with SALMONN which performed poorly on bioacoustics data.
>
> Questions:
>
> > 1. Hard-negative sampling
>
> For thirty percent of prompts, we sample "random" negatives by selecting from all common names or scientific names in our dataset. For the remaining prompts, we choose with uniform probability an ancestor level of either family, order, or phylum, and sample "hard" negatives as species with the same ancestor as the correct species. The number of negatives is chosen uniformly from one to a maximum of thirty-five. We integrate these details into section 3.1.1 for improved clarity.
>
> > 2. XC held-out species
>
> We modified the text in Section 3.2 to clarify that the species were held out at random from the eligible set, which includes species whose genus has at least 100 examples in the dataset as previously specified. We also add the list of all held-out species in the Appendix, and are working towards opening training and benchmarking datasets.
>
> > 3. Music + speech ablation
>
> The results in the paper come from ablating both speech and music together. We include the result of this ablation on all tasks in the Appendix.

---

> > ### Comment · Reviewer_1ReQ · 2024-11-25
> >
> > I thank the authors for addressing my questions and clarifying several points raised in my initial review.
> >
> > This work represents a noteworthy effort to expand recent advancements from general speech (e.g., SALMONN) and audio models (e.g., Pengi) into the bioacoustics domain. However, I believe the contributions fall short of the standards for publication at ICLR. Therefore, I stand by my initial rating and encourage the authors to continue developing and refining this promising line of research.
> >
> > My remaining concerns:
> >
> > 1. Comparisons:
> > Comparing NatureLM-audio to models trained on non-bioacoustic data (e.g., CLAP and SALMONN) in zero-shot settings is unfair.
> >
> > 2. Held-Out Classes:
> > The random selection of held-out species can lead to related species from the same genus appearing in different splits, allowing models to exploit shared patterns. Constraining splits based on taxonomy hierarchy would ensure better separation and more accurate generalization assessment.
> >
> > 3. Evaluation:
> > It is unclear how Nature-LM would perform on recognized bioacoustic benchmarks such as BIRB or BirdSet compared to state-of-the-art models. Evaluation on such benchmarks would provide stronger evidence of the model's generalizability and effectiveness.
> >
> > In its current form, I believe this work would be better suited for a specialized conference or journal, such as ICASSP and JASA, where it could be highly impactful to researchers focused on bioacoustics.

---

> > > ### Author Response · Authors · 2024-11-29
> > > **Concerns about the evaluation**
> > >
> > > We believe that our paper falls into the scope of ICLR: the call for papers includes applications to different domains, representation learning for audio, datasets and benchmarks. We hope that the reviewer considers our contributions on these topics.
> > >
> > > **Comparisons**: We compare against only non-bioacoustic baselines solely on two datasets of the fourteen we evaluate. We do that because the task has not been previously studied and a suitable bioacoustic baseline does not exist. In this case, CLAP and SALMONN are the closest models we can compare with. We can exclude this comparison from the paper if the reviewer thinks it is not informative. We agree stronger baselines for those two tasks would be beneficial, however we note that a first bioacoustic model for these two tasks is one of our contributions.
> > >
> > > **Held-out classes**: Because we want to learn implicit taxonomic relationships that may be useful for unseen species, we sample a held-out dataset of unseen species where the genus is seen. Note that this is already a hierarchical split, chosen for feasible difficulty. In future work, we do intend to study performance when holding out splits at varying levels of the hierarchy. However, the ability to predict unseen species given any split is a novel finding in bioacoustics and a step towards addressing data scarcity in the field with significant implications for the conservation of rare or endangered species.
> > >
> > > **Evaluation**: We chose to evaluate on and extend BEANs which is also an established benchmark in the field and is the most taxonomically diverse, matching our objective. Both BIRB and BirdSet are limited to birds and restrict pre-training data to Xeno-canto. We did compare against SoTA bird models BirdNet and Perch on applicable detection tasks and showed strong performance, but note that outperforming bird models at detection is not a central focus of the paper.
> > > Regarding the comparison on the BirdSet benchmark, we are currently aiming at evaluating NatureLM on these datasets. There are a few issues we encountered:
> > > - The benchmarks’ metrics are more about retrieval than detection. The Top1 accuracy is compatible, but it is not clear whether they include the negative samples (no birds present) along the positives.
> > > - The baselines presented in the benchmark are given the species in each dataset.The classes are known and one-hot encoded whereas this is not true for NatureLM: we decode the responses generatively token by token.
> > > We can release these new datasets and the prompts as part of BEANS-Zero.

---

> ### Author Response · Authors · 2024-11-25
> **Rebuttal feedback**
>
> Dear reviewer 1ReQ, as the discussion phase draws to a close, we look forward to hearing from you. We would be grateful for any thoughts you have on our rebuttal. Thank you!

---

### Official Review · Reviewer_Lfuo · 2024-11-02

**Soundness:** 3
**Presentation:** 2
**Contribution:** 2
**Rating:** 5
**Confidence:** 4

**Summary:**

The authors present a large audio-language model (Lalm) trained and evaluated on bioacoustics data. First, a dataset of text-audio pairs is compiled, partially with the use of LLMs. This dataset is then used to train an architecture similar to SALMONN. It is consisting of a LLM (Llama 3.1-8b), an audio encoder (BEATs), and a Q-Former to connect the computed audio embeddings to the LLM. During training the LLM stays frozen while the audio encoder and Q-Former are trained after a curriculum (first species classification than also detection, captioning, life stage prediction, call-type detection). The model is evaluated on the self introduced BEANS-Zero benchmark which extends BEANS with additional tasks and a zero shot evaluation protocol. The model achieves the best results compared to non bioacoustics

**Strengths:**

1. Addresses a important topic from both the ML research community ( since audio and especially computational bioacoustics is a hard problem) and societal importance.
2. Collects a comprehensive training dataset and extends an existing evaluation benchmark  with additional tasks.
3. The performance improvements compared to a model not trained on bioacoustics data (SALMONN) supports the claim that this domain is in the need for a own foundation model.

**Weaknesses:**

1. Soundness of results: Your presented results only show a minor improvement compared to BioLingual (which also presents zero shot results on BEANS, there numbers differ sometimes why?), so whats the benefit of your approach and more particularly does integrating a LLM has a benefit? Or is it the different training dataset? Or the audio encoder (BEATs vs. HTS-AT)?
2. No further details for replication of the experiments are given, e.g. pretrained models or the list of species which were hold out.
3. Difficult covariate shift from focal training to soundscape test data is not evaluated, which is one of the major challenges in bioacoustics. E.g. for birds see Stowell,2022 or BirdSet,2024.
4. Call-type task: birds usually have more than two call-types per species. A binary classification task ignoring the specie might have few practical applications.
5. L. 428-219: The cbi dataset consists of XC recordings, your model should have the same advantage as BirdNet and Perch, so the comparison should be fair.
6. Line 071-072: How do you support the claim that the generalization of BirdNet is limited? Did not Ghani et al. claim the opposite?

---

### Language

1. l. 190-192 strong repetition
2. Inconsistent use of abbreviations (e.g. state of the art)

**Questions:**

1. Could you add ablation studies to access the influence of each part of your approach? What happens if the audio encoder stays frozen? Are the embeddings that BEATs generates already good enough for taking bioacoustics tasks?
2. What SoTA model is used in the call-type comparison? I could not find that in the text.
3. How do you evaluate the LLM without audio and why is it outperforming your model on the gibbons dataset?

---

> ### Author Response · Authors · 2024-11-21
>
> We would like to thank reviewer Lfuo for the suggestions. We are encouraged that the reviewer appreciated our proposed dataset and the model as the first LALM for bioacoustics.
>
> We address each of the reviewer’s comments below:
>
> Weaknesses:
> > 1. Benefits of using LLMs + Improvement over BioLingual
>
> Using an LLM offers multiple advantages over traditional ML approaches: it can model a wide range of tasks across multiple taxa by leveraging cross-task and cross-taxa similarities, and it can be prompted in natural language and does not require ML expertise or resources for fine-tuning. These features make the model effective in zero-shot scenarios and highly accessible for diverse applications. We integrated an LLM to enable the model to handle tasks like classification, detection, captioning, and individual counting within a single framework in a zero-shot setting, rather than solely to improve performance on specific tasks. Our approach outperforms BioLingual on tasks beyond species classification and detection, and BioLingual cannot handle generative tasks like captioning. Additionally, our detection results differ from the BioLingual paper due to the use of a different metric, as described in Section 4.1.
>
> We are also planning an ablation study to explore the impact of different types of audio encoders; however, this is not the core contribution of the paper.
>
> > 2. More details on the experiments
>
> Regarding the pretrained models, for BEATs we start from [a pretrained BEATs](https://github.com/microsoft/unilm/tree/master/beats) and we fully fine-tune it. For Llama we start from [Llama-3.1-8B-Instruct](https://huggingface.co/meta-llama/Llama-3.1-8B-Instruct) and we fine-tune all attention layers with LoRa. We included these details in Section 4.1.
>
> > 3. Covariate shift
>
> We acknowledge the covariate shift between focal recordings and soundscapes occurring in the detection/retrieval task. While we train on mostly focal recordings, we evaluate on soundscapes, and all the BEANS-Zero detection datasets are soundscapes. We account for the covariate shift by adding environmental noise at random SNR levels between -10 and 10 dB. We have included this info in Section 3.1.1.
>
> Note that NatureLM-audio is still competitive on detection without including a large amount of soundscape datasets (BirdNET and BirdSet), without mixup (all SotA) and using taxonomic losses (Perch). It is very likely that it will improve using the aforementioned changes.
>
> Additionally, we would like to point out that while species detection/retrieval is important, it is only a subfield of bioacoustics, and animal behavior, communication, population structure and other ethology tasks are also important. NatureLM-audio offers the potential to address them under the same framework (see the first point).
>
> > 4. Multiple call types
>
> We created a new evaluation dataset from Xeno-canto classifying between 6 distinct bird call types (*call, song, alarm call, begging call, flight call, and drumming*). On this expanded dataset, our model achieves a performance of **0.686**, significantly better than both the random baseline (~0.167) and BioLingual’s performance (0.339). This could be expanded even further with species-specific calls, however these six types already have different behavioral contexts and should have practical implications for both monitoring and communication studies.
>
> We plan to extend NatureLM-audio to in-context learning which will enable adapting to a wide range of bioacoustics classification tasks defined by (few-shot) examples in prompts.
>
> > 5. Comparison w/ BirdNET and Perch
>
> According to the [BirdNet](https://www.sciencedirect.com/science/article/pii/S1574954121000273) and [Perch](https://www.nature.com/articles/s41598-023-49989-z.epdf) papers the corresponding models are trained on subsets of Xeno-canto *including* the CBI dataset. In addition, BirdNet includes the full Macaulay library and labeled soundscapes around the world. Moreover, [Chasman et al. (2024)](https://openreview.net/forum?id=QCY01LvyKm) identified (footnote at page 8) that BirdNET is likely trained on the Powdermill dataset from which the enabirds is derived. This may explain its high performance on this dataset. Thus, we believe that these frameworks should be considered toplines and not baselines.
>
> We would like to reinforce that we exclude the cbi test set from the xeno-canto training data. We hope that the reviewers will appreciate our fair experimental design.
>
> > 6. Generalization of BirdNET
>
> We have clarified this issue in the manuscript. One important aspect is that BirdNET requires domain adaptation for each of the downstream tasks, while our goal is to have a single *multi-task* foundation model that works across taxa i.e., our model does not require any ML expertise or domain adaptation and can be simply prompted.
>
> Language:  We have verified and corrected the SotA abbreviations in the manuscript, and fixed the strong repetition.

---

> ### Author Response · Authors · 2024-11-21
>
> Questions:
>
> > 1. More ablations BEATs frozen vs unfrozen
>
> We share an ablation training stage-1 model with BEATs frozen vs. unfrozen. We evaluate on classification tasks as the model is trained on only these in stage 1, and because these test most directly the discriminative abilities of the audio encoder.
>
> |   Configuration  | watkins |  cbi  | unseen-cmmn | unseen-sci |
> |----------------------|------------|-------|---------------------|---------------|
> | Beats-Unfrozen|   **0.60**    | **0.58**  |        **0.08**        |       **0.11**      |
> |  Beats-Frozen   |   0.59    | 0.35  |         0.04        |      0.07      |
>
> With gains large on several datasets, this result demonstrates the importance of unfreezing the audio encoder during training and strongly suggest that BEATs embeddings are not sufficient for bioacoustics tasks out-of-the-box.
>
> > 2. Details call-type comparison
>
> This is mentioned in Section 4.4: “We compare against BioLingual for discriminative tasks [..]”. We have also clarified this in the Table 6 caption.
>
> > 3. LLM evaluation without audio
>
> The LLM without audio refers to the original text-based LLM (Llama-3.1-8B-Instruct) that does not receive audio embeddings as part of the input. It is evaluated in the same way as NatureLM-audio. Since it has no information about the audio, its output serves as a random baseline with potential positional bias ([Zheng et al., 2023](https://arxiv.org/abs/2306.05685)). We have clarified this point in the revised paper. This experiment confirms that the gibbons dataset is particularly challenging, and the performance of all models on this dataset falls within random baselines.

---

> ### Author Response · Authors · 2024-11-25
> **Rebuttal feedback**
>
> Dear reviewer Lfuo, as the discussion phase draws to a close, we look forward to hearing from you. We would be grateful for any thoughts you have on our rebuttal. Thank you!

---

> > ### Comment · Reviewer_Lfuo · 2024-11-25
> >
> > Dear authors,
> >
> > Thank you for your comprehensive response to my review. I appreciate the effort you have put into updating the paper.
> >
> > You have addressed most of my concerns, and I have updated my score accordingly.
> >
> > However, I still believe that your contribution lacks some novelty, particularly for machine learning scientists. As I mentioned in my review, it is difficult to discern which aspect contributes to the performance improvement compared to BioLingual. Is it the different dataset, the encoder architecture, the LLM, or the learning procedure?
> >
> > I kindly request that you include the BEATS frozen/unfrozen results in your appendix.

---

> > > ### Author Response · Authors · 2024-11-29
> > > **Performance improvement w.r.t. Biolingual**
> > >
> > > The main advantage with respect to Biolingual is using an actual LLM. We listed the benefits of doing that in the previous reply: NatureLM is a generative model that can handle a wide range of tasks, even zero-shot, you can interact with it using natural text, its performance can scale-up with new tasks, data, size. In terms of contributions, the training data, audio encoder+qformer, and curriculum learning make it possible to leverage a pre-trained LLM like Llama.
> > >
> > > As a concrete example outside of captioning, we refer to the results in Table 6 on call-type and lifestage prediction, as well as the new result on expanded call-types. Our relevant training data uses the same subset of Xeno-Canto used by BioLingual, which already included the lifestage and call type in captions. We nonetheless improve on these tasks significantly, because using an LLM allows us to separate tasks with a unique prompt and response. Large gains in Watkins is another example of this, where long captions appeared to interfere with the species classification for BioLingual but we were able to separate species prediction and captioning through separate prompts.

---

### Official Review · Reviewer_GfHx · 2024-11-08

**Soundness:** 4
**Presentation:** 4
**Contribution:** 3
**Rating:** 8
**Confidence:** 4

**Summary:**

An excellent paper providing a valuable source of bioacoustic data and processing methods to the community. An attempt to comprehensively collect much of the available domain data and curate it into a usable dataset. Plans for distribution and general availability of the dataset/code to the community are missing.

**Strengths:**

An incredible collection of datasets, and careful curation.
A lot of ancillary code for use of the data in various learning tasks.

**Weaknesses:**

Unclear from the presentation if the authors intend to make the dataset widely available, and under what license.

**Questions:**

This is excellent and careful work. I particularly liked the SoTA results in classification

---

> ### Author Response · Authors · 2024-11-20
>
> We thank reviewer GfHx for recognizing the contributions of our paper, particularly the novel datasets (train and benchmark) and the competitive results across multiple tasks. To prevent misuse, we are thoroughly testing the model and carefully determining the best way to make it publicly available. Additionally, we are working towards distributing the training and benchmarking datasets as HuggingFace datasets.

---

> > ### Comment · Reviewer_GfHx · 2024-11-26
> >
> > While I see more defects in the paper, based on the other reviews, I am deciding to maintain my score, as the focus is on curating data and baseline algorithms to advance the field. I wish the authors luck, and I do hope the data will be made widely available.

---

### Author Response · Authors · 2024-11-20
**Corrections to enabirds and dcase-bird results in Table 4**

After re-running the evaluation and verifying results, we found out that the results for BirdNET and Perch for enabirds in Table 4 were interchanged. The detection accuracy for BirdNET is 0.49 while Perch is 0.164. This result is consistent with recent findings in [Chasman et al. (2024)](https://openreview.net/forum?id=QCY01LvyKm): BirdNET high performance on this dataset may be explained by the fact that they use Powdermill as their training data which overlaps with enabirds. We additionally update NatureLM-audio’s result in the DCASE-bird dataset in Table 4 from 0.052 to 0.088 after limiting the dataset to bird classes to match the BirdNet and Perch evaluation setting. We would like to apologize for these errors which we corrected in the paper.

---

### Meta-Review · Area_Chair_pfJ4 · 2024-12-22

**Metareview:**

**Paper Summary:**

This paper describes an audio-language foundation model for bioacoustics: NatureLM. This model is trained on a new collection of annotated datasets consisting of bioacoustics, speech, and music. It is evaluated using the newly proposed BEANS-Zero benchmark.

**Strengths:**

Reviewers GfHx and Lfuo both appreciate the value of the training data collected to train this model. In discussion, the authors clarified that:

> We will publish the datasets on HuggingFace [...] This will be the largest open cross-taxa multi-task bioacoustics dataset.

Reviewers GfHx and 1ReQ both appreciate the development of the BEANS-Zero benchmark.

**Weaknesses:**

Reviewers Lfuo and 1ReQ raised several concerns about the experimental results, most of which were satisfactorily addressed in the discussion period.

**Additional Comments On Reviewer Discussion:**

The main unresolved concern raised by Reviewers Lfuo and 1ReQ in the discussion period is whether NatureLM is in-scope for ICLR. Reviewer Lfuo is concerned about lack of novelty, from the perspective of machine learning (clearly there is novelty in the data and benchmarking contributions to bioacoustic modeling). Reviewer 1ReQ believes that "this work would be better suited for a specialized conference or journal, such as ICASSP and JASA, where it could be highly impactful to researchers focused on bioacoustics.

I tend to disagree with this concern: machine learning research has always welcomed and benefited from work on applications. If anything, I think that contributions to a less crowded application like bioacoustics have greater value than the more common vision/NLP applications that are always welcome at ICLR.

---

### Decision · Program_Chairs · 2025-01-22

Accept (Poster)